# Development of Methods Derived from Iodine-Induced Specific Cleavage for Identification and Quantitation of DNA Phosphorothioate Modifications

**DOI:** 10.3390/biom10111491

**Published:** 2020-10-28

**Authors:** Sucheng Zhu, Tao Zheng, Lingxin Kong, Jinli Li, Bo Cao, Michael S. DeMott, Yihua Sun, Ying Chen, Zixin Deng, Peter C. Dedon, Delin You

**Affiliations:** 1State Key Laboratory of Microbial Metabolism, Joint International Research Laboratory of Metabolic and Developmental Sciences, and School of Life Sciences & Biotechnology, Shanghai Jiao Tong University, Shanghai 200030, China; ecnubio32@163.com (S.Z.); ashlyzheng@hotmail.com (T.Z.); konglingxin7@163.com (L.K.); elisalijinli@163.com (J.L.); 13162732720@163.com (Y.S.); chenying2014@tju.edu.cn (Y.C.); zxdeng@sjtu.edu.cn (Z.D.); 2College of Life Sciences, Qufu Normal University, Qufu 273165, Shandong, China; caobogxu@126.com; 3Department of Biological Engineering and Center for Environmental Health Science, Massachusetts Institute of Technology, Cambridge, MA 02139, USA; msdemott@mit.edu (M.S.D.); pcdedon@mit.edu (P.C.D.); 4Antimicrobial Resistance Interdisciplinary Research Group, Singapore-MIT Alliance for Research and Technology, 1 CREATE Way, Singapore 138602, Singapore

**Keywords:** DNA, phosphorothioate (PT) modifications, iodine-induced cleavage (ICA), ICDS, PT-IC-Seq

## Abstract

DNA phosphorothioate (PT) modification is a novel modification that occurs on the DNA backbone, which refers to a non-bridging phosphate oxygen replaced by sulfur. This exclusive DNA modification widely distributes in bacteria but has not been found in eukaryotes to date. PT modification renders DNA nuclease tolerance and serves as a constitute element of bacterial restriction–modification (R–M) defensive system and more biological functions are awaiting exploration. Identification and quantification of the bacterial PT modifications are thus critical to better understanding their biological functions. This work describes three detailed methods derived from iodine-induced specific cleavage-an iodine-induced cleavage assay (ICA), a deep sequencing of iodine-induced cleavage at PT site (ICDS) and an iodine-induced cleavage PT sequencing (PT-IC-Seq)-for the investigation of PT modifications. Using these approaches, we have identified the presence of PT modifications and quantized the frequency of PT modifications in bacteria. These characterizations contributed to the high-resolution genomic mapping of PT modifications, in which the distribution of PT modification sites on the genome was marked accurately and the frequency of the specific modified sites was reliably obtained. Here, we provide time-saving and less labor-consuming methods for both of qualitative and quantitative analysis of genomic PT modifications. The application of these methodologies will offer great potential for better understanding the biology of the PT modifications and open the door to future further systematical study.

## 1. Introduction

Naturally occurring epigenetic modifications of DNA have been found prevalent in organisms from all domains of life [1]. Usually, those modifications involve a variety of chemical groups (such as methyl group, amino acids, polyamines, monosaccharides, and disaccharides) appended to the nucleobase portion of a nucleotide. These modifications do not alter the specificity of base pairing but play an important role in protection and genetic regulation. DNA phosphorothioate (PT) modifications, in which the non-bridging phosphate oxygen is replaced by sulfur, occurs at the sugar-phosphate backbone rather than the nucleobase in a sequence- and *R*p stereo-specific manner [2,3,4,5].

The discovery of natural PT modifications originated from the DNA degradation (Dnd) phenotype during the electrophoresis of genomic DNA (gDNA) isolated from *streptomyces lividans* 1326 [6]. Since then, the Dnd phenotype has been considered to be a typical method for detection of PT modification. This modification turned out to be the sulfur-for-oxygen substitution supported by the isotope incorporation during mass spectrometric analysis and all identified physiological PT internucleotide linkages were in the unique stereochemistry of *R*p configuration [7]. PT modification made DNA susceptible to oxidative cleavage by a peracid derivative of Tris formed at the electrophoretic anode [8]. Meanwhile, the modified DNA possessed enhanced resistance to nuclease (like nuclease P1), which set the stage for the generation of PT-linked dinucleotides facilitating the quantification of genomic PT modification by liquid chromatography-mass spectrometry (LC-MS) [3]. Further studies have demonstrated that this modification is governed by the *dndABCDE* cluster in *streptomyces lividans* 1326 [7], and deleting of *dnd* cluster could abolish the modification [9]. PT modification has been found in many other bacteria and the functional study of the *dnd* clusters revealed the necessity of the five proteins [10]. DndA is regarded as a pyridoxal 5′-phosphate-dependent cysteine desulfurase and supplying sulfur for PT modification, DndC exhibited homology to 3′-phosphoadenosine-5′-phosphosulphate (PAPS) reductase and showed adenosine-triphosphate (ATP) pyrophosphatase activity in vitro [11], DndD carried the conserved ATP-binding motif and was suggested to provide energy for stabilizing DNA or site-specific DNA nicking [12], and DndE has been shown to bind the nicked double-stranded DNA substrates preferentially in a sequence-independent manner [13,14]. DndB negatively regulated the PT modification as the deletion of dndB aggravated DNA degradation and increased PT frequencies [15,16]. The biochemical study of PT modification revealed that PT-modifying enzymes DndACDE function as a large protein complex [17]. However, it is still poorly understood how the Dnd proteins incorporate the sulfur into DNA synergistically. Currently, PT modifications have been found in over 1000 bacterial and archaeal strains but not in eukaryotes, and their biological functions are still poorly understood [10]. In some bacteria, the *dndABCDE* contribute to the constitution of the defensive R–M system with *dndFGH*, by which the PT-carrying host could distinguish and attack non-PT-modified foreign DNA [18]. Other biological functions of PT modification remain to be explored, which require more characterization of PT modification in a variety of bacteria. To this end, quantitative analysis of PT modifications and whole genomic landscape descriptions of the location and frequency of PT modification are thus important in bacteria. The tolerance of PT-modified DNA enabled the sequence identification and frequency quantitation by LC-MS/MS in the two-nucleotide sequence context after hydrolysis of the DNA samples by nucleases [3]; however, this method is complicated and labor-intensive.

Long before the discovery of natural PT modifications, the modification was developed by artificial chemical approaches to protect oligodeoxynucleotides against nuclease degradation [19,20]. Previous works showed that compared with the native form of DNA, the PT-modified DNA has special chemical properties and is susceptible to cleavage by some chemical reagents, such as 2-iodoethanol [21]. Subsequently, it found that three products-3′- or 5′-fracture phosphate DNA-and desulfurization produce normal phosphate two-ester-bond DNA, and are produced through selective cleavage of PT linkages by 2-iodoethanol [22]. Moreover, the higher cutting efficiency (nearly 100%) for the DNA containing PT modifications could be achieved under the optimized reaction system [23,24,25,26]. Taking advantage of these special properties, an iodine-induced cleavage assay was developed to rapidly determine whether there is PT modification on the DNA. This method is easy to follow compared with LC-MS/MS and has become the most common method used for PT modification discovery [27,28,29,30]. As PT modification is a partial and highly dynamic modification, more techniques should be developed to locate the PT modification sites and uncover the diverse sequence selectivity and quantity in different bacteria, even in an individual cell of a population, because of the heterogeneity of PT modification [27,31].

Here, we describe a series of methods based on PT-specific iodine-induced cleavage and high-throughput next-generation sequencing technologies: ICA, ICDS, and PT-IC-Seq [27,31] (See Figure 1 for a schematic overview).

Among them, the ICA, a method of rapidly detecting PT modification status in the genome, has been developed by using an electrophoresis pattern of the iodine-cleaved gDNA. The DNA samples were cleaved by iodine solution under an optimized condition, and the presence of PT modifications was first detected by electrophoresis in agarose gel, and then confirmed by LC-MS/MS. The ICDS technology combining high-throughput Illumina sequencing with iodine-specific cleavage maps the locating of PT-modified sites in the genome. The PT-modified sites were cleaved specifically by iodine, and the resultant cutting sites were then marked with special labels. The tag location sites could be detected by the two-generation sequencing, which in turn fulfilled the determination of the distribution and localization of genomic PT modification sites. The PT-IC-Seq technology was an updated derivative of ICDS with higher throughput and resolutions. For the quantitative determination of PT modification percentage at each site, the DNA samples were first cleaved by iodine at selected phosphorothioate linkages, followed by tag ligations and high-throughput sequencing. No enrichment process was required for mapping sequenced reads to the reference genome. Taken together, these methods could be applied to investigate and quantitatively analyze PT modifications at whole genomic landscapes, and will expand our knowledge of PT modifications in the future.

## 2. Materials and Methods

### 2.1. Materials

*E. coli* strain DH10B was purchased from Novagen. *Salmonella enterica* serovar Cerro 87 was supplied by Professor Toshiyuki Murase (Tottori University, Japan), and *E. coli* strain B7A was obtained from Dr Jaquelyn Fleckenstein (Departments of Medicine and Molecular Sciences, University of Tennessee Health Science Center).Mono nucleosides standard samples (A, G, C, T) and PT-modified dinucleotide standard samples (GpsA, GpsT) were synthesized by Sangon Biotech Co. Ltd. (Shanghai, China).MicroSpin G-25 columns (GE Healthcare, Pittsburgh, PA, USA).Pipettes, tips and microcentrifuge tubes (Eppendorf, Hamburg, Germany).

### 2.2. Reagents

Ethanol (Sigma-Aldrich, St. Louis, MO, USA).Acetic acid (Sigma-Aldrich, St. Louis, MO, USA).I_2_ (Sigma-Aldrich, St. Louis, MO, USA).dNTP (Sigma-Aldrich, St. Louis, MO, USA).Nuclease P1 (Sigma-Aldrich, St. Louis, MO, USA).DNA ladder (Thermo Fisher Scientific, Waltham, MA, USA).DNA Gel Loading Dye (Thermo Fisher Scientific, Waltham, MA, USA).Fast AP Thermosensitive Alkaline Phosphatase (Thermo Fisher Scientific, Waltham, MA, USA).Klenow Fragment (3′→5′ exo-) (New England BioLabs, Ipswich, MA, USA).Buffer 2 (New England BioLabs, Ipswich, MA, USA).dATP (New England BioLabs, Ipswich, MA, USA).T4 DNA ligase (New England BioLabs, Ipswich, MA, USA).

### 2.3. Chemicals

Agarose (Sigma-Aldrich, St. Louis, MO, USA).NaCl (Sigma-Aldrich, St. Louis, MO, USA).Na_2_HPO_4_ (Sigma-Aldrich, St. Louis, MO, USA).EDTA (Sigma-Aldrich, St. Louis, MO, USA).ZnCl_2_ (Sigma-Aldrich, St. Louis, MO, USA).Thiourea (Sigma-Aldrich, St. Louis, MO, USA).Sodium acetate (Sigma-Aldrich, St. Louis, MO, USA).Tryptone (Sigma-Aldrich, St. Louis, MO, USA).Yeast Extract (Sigma-Aldrich, St. Louis, MO, USA).Tris (Sigma-Aldrich, St. Louis, MO, USA).

### 2.4. Kits

Bacterial DNA Kit (OMEGA, Norcross, GA, USA).Plasmid Mini Kit (OMEGA, Norcross, GA, USA).Bacterial DNA Kit (OMEGA, Norcross, GA, USA).Cycle Pure Kit (OMEGA, Norcross, GA, USA).Quant-iT™ PicoGreen™ dsDNA Assay Kit (Thermo Fisher Scientific, Waltham, MA, USA).Quick BluntingTM kit (New England BioLabs, Ipswich, MA, USA).NEBNext DNA Library Prep Reagent Set for Illumina (New England BioLabs, Ipswich, MA, USA).EquipmentHorizontal Electrophoresis Systems (BIO-RAD, Hercules, CA, USA).T100TM Polymerase Chain Reaction (PCR) Systems (BIO-RAD, Hercules, CA, USA).1260 Infinity II High Performance Liquid Chromatography (HPLC) Systems (Agilent, Palo Alto, CA, USA).QQQ 6470 LC-MS/MS Systems (Agilent, Palo Alto, CA, USA).Bioanalyzer Instrument 2100 (Agilent, Palo Alto, CA, USA).Heated Circulating Baths (Thermo Fisher Scientific, Waltham, MA, USA).NanoDrop 2000 (Thermo Fisher Scientific, Waltham, MA, USA).Qubit™4 Fluorometer (Thermo Fisher Scientific, Waltham, MA, USA).UV Spectrophotometer (HACH, Loveland, CO, USA).Constant Temperature Heater (Eppendorf, Hamburg, Germany).Sonoplus HD2070 Sonicator (Bandelin, Berlin, Germany).SPRIworks Fragment Library System (Beckman, Brea, CA, USA).Freeze Dryer (Labconco, Kansas City, MO, USA).

### 2.5. Primers

Duplex tag-sequenceF1: 5‘-Phos/TTTAACCGCGAATTCCAG/ddC/-3′ (DNA, HPLC-purified)R1: 5‘-GCTGGAATTCGCGGTTAAAT-3′ (DNA, HPLC-purified)Enriched PCR primersF2: 5‘-CAAGCAGAAGACGGAATACGA-3′ (DNA, PAGE-purified)R2: 5‘-AATGATACGGCGACCACCGAGATCTACACTCTTTCCCTACACGACGCTCTTCCGATCTGCTGGAATTCGCGGTTAAAT-3′ (DNA, PAGE-purified)

### 2.6. Methods

#### 2.6.1. Sample Preparation (Using *E. coli* B7A as an Example)

Isolate a single colony from a freshly streaked selective plate and regrown in Luria-Bertani (LB) medium at 37 °C for overnight or to logarithmic phase.Harvest cells by centrifugation at 10,000 rpm for 2 min.Aspirate and discard the media.Extract gDNA using the Bacterial DNA Kit according to the manufacturer’s instructions.Store eluted DNA at −80 °C until ready for use in following experiments.[Note: For optimal DNA yields, the starting culture volume should be based on culture cell density. From experience, the cell density at 600 nm (OD600) of 2.0~3.0 is recommended.]

#### 2.6.2. ICA Assay

Prepare the 10 × ICA Reaction Buffer (500 mM Na_2_HPO_4_, pH 9.0, 30 mM iodine).Add ~1 μg gDNA with 1 × ICA Reaction Buffer to clean microcentrifuge tube. Mix well, and incubate at 65 °C for 10 min and then slow cooled (0.1 °C/s) to 4 °C using a thermal cycler.Analyze reaction sample using ethidium bromide-stained 1.0% agarose gel electrophoresis in 1 × TAE buffer containing 50 μM thiourea.At the same time, analyze the above samples using the LC-MS/MS method (see Quantification of PT modifications in DNA by LC-MS/MS section).[Note: Always prepare a fresh iodine for every use, and confirm slow cooling process. Reduce voltage and increase gel running times possible during electrophoresis. These tips may yield better results.]

#### 2.6.3. Establishment of Standard Curve

The Standard Curve of Mono Nucleosides

Dilute the mono nucleosides standards (C and T) to six different concentrations (0 ng/μL, 2 ng/μL, 4 ng/μL, 10 ng/μL, 20 ng/μL, 40 ng/μL).[Note: The total number of A is equal to the number of T, and similarly the number of C is equal to the number of G. Thus, we could choose two mono nucleosides to draw the standard curve.]Analyze samples using the HPLC system with an Agilent ZORBAX SB-C18 column (3.5 μm, 2.1 × 150 mm), and the conditions as follow:Solvent A was 0.1% acetic acid in water and Solvent B was 0.1% acetic acid in acetonitrile; the gradient program was 0–3 min 0.5% B, 3–4 min 0.5–5% B, 4–15 min 5–12% B, 15–16 min 12–30% B, 16–20 min 30% B, 20–21 min 30–0.5% B, and 21–25 min 0.5% B; the flow rate was 0.3 mL/min with elution conducted at 35 °C.Analyze the results: with the concentration of mono nucleosides standards (ng/μL) as the abscissa (X) and the peak area values of HPLC as the ordinate (Y), the standard curve was drawn.[Note: Always prepare a fresh set of standards and record the samples volume for every use.]

The Standard Curve of Dinucleotides

Dilute the *R*p configuration of the PT-modified dinucleotide standards to seven different concentrations (0 fmol/μL, 2 fmol/μL, 10 fmol/μL, 20 fmol/μL, 40 fmol/μL, 100 fmol/μL, 200 fmol/μL).Add ~100 fmol/μL the *S*p configuration of PT-modified dinucleotides to each standard used as internal reference.Analyze samples using the LC-MS/MS (see section Quantification of PT modifications in DNA by LC-MS/MS), and the analysis of results as follow:With the concentration of *R*p configuration of the PT-modified dinucleotide (fmol/μL) as the abscissa (X) and the ratio of the standard peak area to internal reference peak area values measured by LC-MS/MS as the ordinate (Y), the standard curve was drawn.[Note: Always prepare a fresh set of standards and record the samples volume for every use.]

#### 2.6.4. Quantification of PT Modifications in DNA by LC-MS/MS

Prepare the nuclease P1 Hydrolysis Buffer (0.5 mM ZnCl_2_, 30 mM sodium acetate, pH 5.3).Add ~20 μg purified DNA, 2 U of nuclease P1 and Hydrolysis Buffer to a clean microcentrifuge tube. Mix well, and incubate at 50 °C for 2 h using a thermal cycler.Then mix in 100 mM Tris-HCl, pH 8.0 and 15 U of FastAP Alkaline Phosphatase, and incubate at 37 °C for 2 h for the totally dephosphorylation.Remove the enzymes by ultrafiltration at 10,000 rpm for 10 min.[Note: This step is critical for removal of enzymes that may interfere with follow-up experiments.]Add ~200 μL deionized water to ultrafiltration tube and repeat step 4 until all the sample has been transferred to the collection tube.Collect the filtrate in a clean microcentrifuge tube and dry the sample.[Note: For subsequent LC-MS/MS analysis, the volume of sample is as small as possible. Concentrated sample using the Freeze Dryer may yield better results.]Resuspend and elute the sample from step 6 in 40 μL deionized water.Analyze sample quantitatively by LC-MS/MS system with an electrospray ionization source in positive mode as described previously [2], and the parameters as follow:Gas flow, 10 L/min; nebulizer pressure, 30 psi; drying gas temperature, 325 °C; and capillary voltage, 3100 V. Multiple reaction monitoring mode was used for detection of product ions derived from the precursor ions, with all instrument parameters optimized for maximal sensitivity (retention time in min, precursor ion m/z, product ion m/z, fragmentor voltage, collision energy): GpsA, 9.4, 597, 136, 120 V, 40 V; GpsT, 11.2, 588, 152, 110 V, 17 V.Analyze the results according to standard curves: the hydrolyzed mono nucleotides were quantified by HPLC according to the standard curve of mono nucleosides. Meanwhile, the PT-modified dinucleotides were quantified by LC-MS/MS according to the standard curve of dinucleotides. Thus, the number of PT-modified dinucleotides in a unit length of DNA can be calculated.

#### 2.6.5. ICDS Assay

Cleave ~20 μg purified gDNA by iodine as described above in ICA methods.Remove residual iodine and salts using MicroSpin G-25 columns.[Note: This step is critical for removal of residual iodine and salts that may interfere with follow-up experiments.]Add 10 U of FastAP Alkaline Phosphatase to the sample at 37 °C for 1h for remove terminal phosphate groups.Inactivate the enzyme by heating at 75 °C for 5 min and slow cooled (0.1 °C/s) to 4 °C to assure proper complementary re-annealing.Clean up the sample using the Cycle Pure Kit and elute it in 30 μL MilliQ water.Blunt end of the break sites using the Quick BluntingTM Kit at room temperature for 30 min.Inactivate the enzyme by heating at 75 °C for 10 min and slow cooled as before.Clean up and elute the sample as before.Then mix in 1x NEB Buffer 2, 0.1 mM dATP and 15 U of Klenow Fragment (3′→5′ exo-) and incubate at 37 °C for 30 min for the 3′-deoxyadenylation (namely A-tailing).Inactivate the enzyme by heating at 75 °C for 20 min and slow cooled as before.Clean up and elute the sample as before.Combine 3 μM of custom duplex tag-sequence (see Primers section) with 3′-deoxyadenylated ends by T4 DNA ligase at 16 °C for 16 h.Inactivate the enzyme by heating at 75 °C for 10 min and slow cooled as before.Clean up and elute the sample as before.[Note: The quality of the DNA samples was measured using NanoDrop 2000 and their concentration was measured using the Quant-iT™ PicoGreen™ dsDNA Assay Kit.]Shear the DNA samples to fragment lengths between 150 and 350 bp by sonication, and then ligate standard Illumina sequencing adaptor using the SPRIworks Fragment Library System.[Note: To fragment the DNA samples to a size range of 150–350 bp, using a probe sonicator at an amplitude of 20% with 20 s “ON” and 10 s “OFF” (10 min total), while on ice to avoid excessive heat.]Concentrate samples using the Freeze Dryer and elute it in 30 μL MilliQ water.Enrich the iodine-cleaved DNA linked unique tag by PCR amplification for 15 cycles using F2 and R2 as the primers.[Note: During the PCR amplification, one of the primers used matches the marked segments attached to the ends of cleavage by iodine (F2), while the other matches the standard Illumina sequencing adaptor (R2). This allows only fragments containing the iodine-cleaved ends to be amplified, which achieves the enrichment of the PT-modified molecules.]Sequence the libraries constructed from step 17 on the Illumina HiSeq X Ten platform.[Note: Agilent Technologies 2100 Bioanalyzer is used to confirm successful library generation and Life Technologies Qubit 3.0 Fluorometer for quantification.]Analyze the ICDS sequencing data, and the details as follow:After completing Illumina sequencing, the reads containing tag were trimmed for adaptor and tag and done quality control as follows:
Clipping the adapter sequences.Removing non-A, G, C, T bases of the 5′ end.Trimming low-quality base (less than Q20).Removing reads with more than 10% of “N” calls.Filtering small fragments with less than 25 bp after clipping the adapter sequences and quality control, and then aligned to reference genomes by Burrows-Wheeler Aligner (BWA) and the position-wise coverage values were calculated by using the custom python script. The GAAC/GTTC sites will be defined as PT-modified sites if their reads above 50 and ended at this site. Meanwhile, 10 non-GAAC/GTTC sites were randomly selected as control.

#### 2.6.6. PT-IC-Seq Assay

Cleave ~20 μg purified gDNA by iodine as described above in ICA methods.Remove residual iodine and salts using MicroSpin G-25 columns.[Note: This step is critical for removal of residual iodine and salts that may interfere with follow-up experiments.]Shear the DNA samples to fragment lengths of 150–350 bp by sonication.[Note: To fragment the DNA samples to a size range of 150–350 bp, using a probe sonicator as before.]According to instructions provided with the NEBNext DNA Library Prep Reagent Set for Illumina, the resulting fragments were end-repaired, adenylated at the 3′ ends and ligated to Illumina paired-end adaptors.Amplify DNA fragments by PCR for 15 cycles using standard Illumina adapter-specific primers.Sequence the libraries constructed from step 5 on the Illumina HiSeq X Ten platform.[Note: Agilent Technologies 2100 Bioanalyzer is used to confirm successful library generation and Life Technologies Qubit 3.0 Fluorometer for quantification.]Analyze the PT-IC-Seq data, and the details as follow:All sequencing reads were trimmed for adaptor and low-quality bases and aligned to reference genomes by BWA for creating Sequence Alignment/Map (SAM) files. Then, SAM files were converted to Binary Alignment/Map (BAM) files that were piled up using samtools and the results were visually performed using the Integrated Genomics Viewer 2.3 software (IGV; Broad Institute, Cambridge, MA, USA). Meanwhile, the position-wise reads number obtained from fragment terminals and across the same site were calculated respectively by using the custom python script. The PT modification frequency of each GAAC/GTTC sites were calculated by the number of reads ended at this site divided by all of the number of reads ended and crossed the same sites. To eliminate the false-ended reads arising from random shearing the DNA, 10 non-GAAC/GTTC sites used as the control and their average PT modification frequency were calculated and used them as thresholds. The modification ratios at each PT sites in the whole genome will be calculated and analyzed if the modification frequency of those sites were above the set thresholds.

## 3. Results

### 3.1. ICA Assay

For the feasibility assessment of ICA method, the gDNA isolated from *E. coli* B7A containing PT-modified genes, was tested as an example. The ICA assay showed that treatment of the PT-modified gDNA with iodine resulted in smaller fragment distribution compared with the control treated without iodine, whereas the gDNA from *E. coli* DH10B lacking PT gene cluster was not cleaved (Figure 2A).

The results demonstrated that iodine could cleave PT-modified DNA with high efficiency. This finding was consistent with the previous cleavage studies [20,21,22,23], so this assay is feasible to detect the presence of PT modifications in the genome. To consolidate the above results, we re-analyzed PT modifications in gDNA of *E. coli* B7A by LC-MS/MS. As shown in Figure 2B,C, the PT-containing dinucleotides (GpsA or GpsT) after hydrolysis of the gDNA by nucleases is consistent with the retention time of the standard that possessed PT modifications with *R*p configuration. These results confirmed that the ICA approach could rapidly detect PT modification status in the genome.

### 3.2. Establishment of Standard Curve

According to above the method, the standard curves of different mono nucleosides standards are shown in Appendix A for T and Appendix A for C, and the standard curves of different PT-modified dinucleotide standards are shown in Appendix A (GpsA) and Appendix A (GpsT).

### 3.3. Quantification of PT Modifications in DNA by LC-MS/MS

According to above the method, the dinucleotides of gDNA isolated from *E. coli* B7A were quantitively analyzed as an example. By using synthetic mono nucleosides and PT-modified dinucleotide standards, the amount of PT-containing dinucleotides after hydrolysis of the gDNA by nucleases was analyzed by LC-MS/MS (Figure 3A,B) and HPLC (Figure 3C).

We were thus able to detect PT modification levels occurring in GpsA and GpsT contexts at 325 ± 8 and 361 ± 11 PTs per 10^6^ nt, respectively in 20 μg gDNA (Figure 3D), which is consistent with previous research [3].

### 3.4. ICDS Assay

The ICDS method was used to characterize the PT-modified on the gDNA isolated from *E. coli* B7A, which was tested as an example. By aligning the output reads to the reference genome sequence and visual analysis using the IGV, the partial data were visually depicted in Figure 4A, and the GAAC/GTTC sites in *E. coli* B7A genome were defined as PT-modified sites, with their reads above 50 with the unique tag and ending at this site (such as the red triangle mark in Figure 4A).

The distribution and localization of *E. coli* B7A genome PT modification sites were determined and shown in Figure 4B. These results demonstrated that the method was feasible for discerning PT sites in the whole genome and can draw the high-resolution genomic maps of PT modifications. However, this approach was not applied in single-stranded modifications, such as *Vibrio cyclitrophicus* FF75 [27].

### 3.5. PT-IC-Seq Assay

As the PT modifications have been enriched during the PCR amplification step, the ICDS method can only be used for determining PT sites, not for quantification. Taking the limits of ICDS method into consideration, a novel approach named PT-IC-Seq was developed.

PT-IC-Seq method was applied to quantify the PT modification on the gDNA isolated from *Salmonella enterica* serovar Cerro 87, which was tested as an example. By aligning the output reads to the reference genome sequence and visual analysis using the IGV, the partial data were visually depicted in Figure 4C. There are two main forms at the GAAC/GTTC site. One is that the intact GAAC/GTTC motifs appeared at the internal of sequencing reads, which originated from unmodified GAAC/GTTC sites, and the other is initiated with AAC or TTC, regarded as PT-modified sites, which resulted from the cleavage of the DNA strand at the PT-modified GpsAAC/GpsTTC site by iodine. The PT modification level at each modified site throughout the *Salmonella enterica* serovar Cerro 87 genome was calculated (Appendix A) and shown in the PT modification map (Figure 4D). These results demonstrated the method was feasible for quantitatively characterizing the genomic landscape of PT modifications and determining PT modification frequency at each modified site.

## 4. Discussion

The nucleic acids contain diverse chemical modifications, which exerts important influence in a variety of life processes [32,33,34]. Among these modifications, DNA methylation is the best known and has essential roles in cellular processes, such as genome regulation, development, and disease [35,36,37]. The recently discovered PT modifications occurring on the DNA backbone was a novel DNA modification, in which the non-bridging phosphate oxygen is replaced by sulfur. This exclusive DNA modification is widespread in bacteria within a sequence selective and *R*p stereo-specific manner, but not found in mammal cells, and their physiological functions remain poorly understood. Thus, determination and quantitative analysis of the PT modifications are essential for exploring and understanding their biological functions. We provide here a comprehensive description for the entire process of a series of approaches for analyzing PT modifications in the genome so that it can be easily adopted by researchers.

Since the initial report on PT modification detection using the ICA assay in 2014 [27], the technique has been widely used for PT modification discovery. An important advantage of this method is its simplicity, because the detection of PT modifications can be completed in a relatively short time and at low cost. The main goal of the ICA method is to rapidly detect the presence of PT modifications in the genome. The treatment of iodine introduced a strand break at the PT-modified site, providing the basis for new technology development. During the development of the method, we found that the quality of iodine was the key to successful experimentation according to previous results. From experience, a fresh iodine solution should be prepared every time. Meanwhile, the reduction of voltage and extending gel running time during electrophoresis process of ICA assay may help to yield better results. Importantly, we also used the LC-MS/MS method to further confirm the results after the ICA assay, which ensure the accuracy of the results.

To further understand the characteristics of PT modifications in the context of the genomic landscape, we developed two highly novel approaches: ICDS and PT-IC-Seq, which integrate the integration of PT-specific iodine-induced cleavage with high-throughput next-generation sequencing technology. As illustrated in Figure 1, ICDS was developed to map PT locations in bacterial genomes based on the adaptation of high-throughput next-generation sequencing technology. In the ICDS approach, iodine reagent was introduced to cleave DNA at PT modification sites, and then ligated to an adaptor with a specific index sequence for enriching the DNA fragments with PT modifications by PCR amplification. Since the PCR amplification step will result in enriched amplicons with PT modifications, ICDS sequencing approach can only be used for identifying PT sites but not for quantification. PT-IC-seq was then developed to quantitatively determine the PT modification percentage at each site without enrichment process. The PT-modified motifs would be cleaved and should be presented as the reads ends in the sequencing output. The unmodified motifs would not be cleaved and should be present in the internal locations of DNA fragments. The frequency of PT modification can be determined by the ratio of a sequence reads at the end versus internal, so the modification was quantified authentically. However, ICDS sequencing could not do this. Moreover, the sites with lower modification frequency missing in ICDS could also be detected by PT-IC-Seq with high sequencing depth. Compared with the conventional methods such as Dnd phenotype electrophoresis analysis and LC-MS/MS, ICDS and PT-IC-Seq technology were convenient and efficient approaches to characterize PT modification in the context of the genome-wide scale quantitatively and accurately. Unfortunately, neither the ICDS nor the PT-IC-Seq method can be used to analyze strains possessing single-stranded PT modifications, such as *Vibrio cyclitrophicus* FF75. In fact, when the PT modification occurs on a single strand, the treatment of iodine could not introduce a double-strand break at modified sites. Therefore, ICDS and PT-IC-Seq methods are bistranded PT modification-specific methods. Even so, this limitation may not impact the application for PT modifications research, because almost all reported PT modifications have occurred on the double-strand of DNA to date, except for a few bacterial such as the *Vibrio cyclitrophicus* FF75. However, these methods set important stages for new derivative methods, such as iodine-induced cleavage quantitative real-time PCR (IC-qPCR), which integrated the iodine-induced cleavage and the TaqMan real-time PCR, and was used to assess the frequency of PT modification at a given site in bacterial genome [38]. Other methods may be developed in the future for the study of PT modification.

## 5. Conclusions

In summary, we describe a series of methods coupling PT-specific iodine-induced cleavage with high-throughput next-generation sequencing technologies, to identify and quantify PT modifications in the genome. Using the ICA method, we have achieved rapid detection of PT modification status in the genome. Moreover, to characterize the PT modification on whole genomic landscapes, ICDS technology was developed. We have successfully applied it to identify features of this epigenetic mark at any one of genomic positions. Based on the success, a high-resolution map of locations for PT-modified sites has been achieved. Furthermore, the PT-IC-Seq technology was developed on the basis of the ICDS method, which was able to quantitatively analyze the frequency of specific PT-modified sites on the genome. Overall, the protocols not only pave a path to a better understanding of the biology of the PT modification, but also serve as a useful technique suitable for the investigation of PT modification related studies.

## Figures and Tables

**Figure 1 biomolecules-10-01491-f001:**
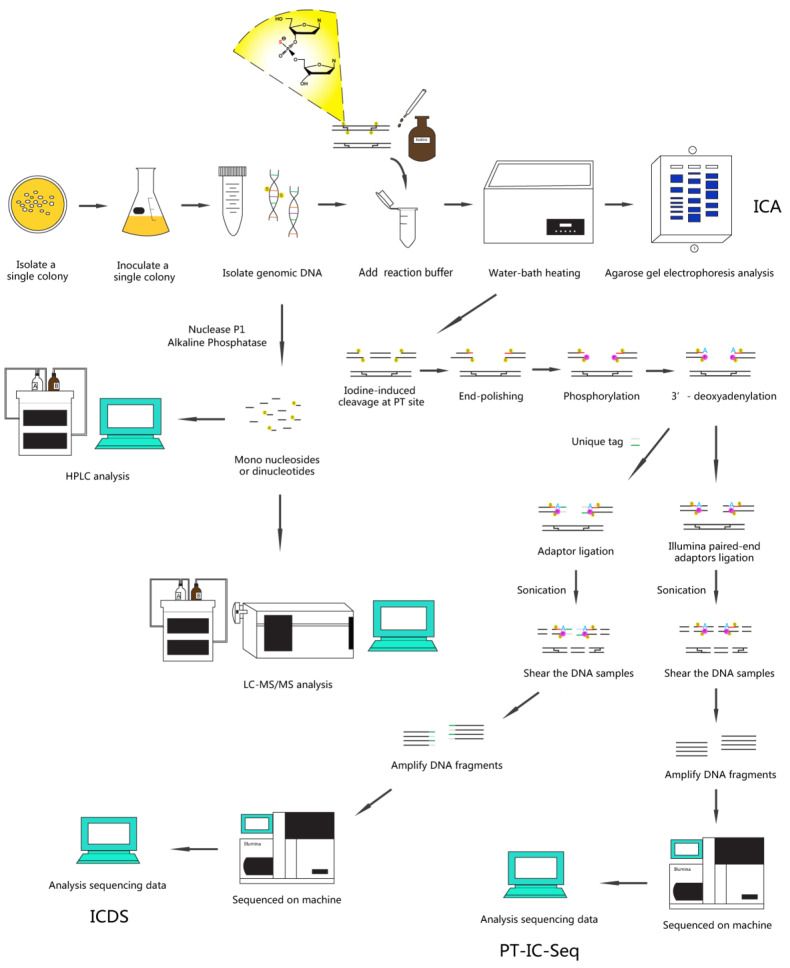
Overview of the PT-specific iodine-induced cleavage-related methods. Depiction of the key steps in the three experimental protocols from PT modifications identification to quantitative analysis.

**Figure 2 biomolecules-10-01491-f002:**
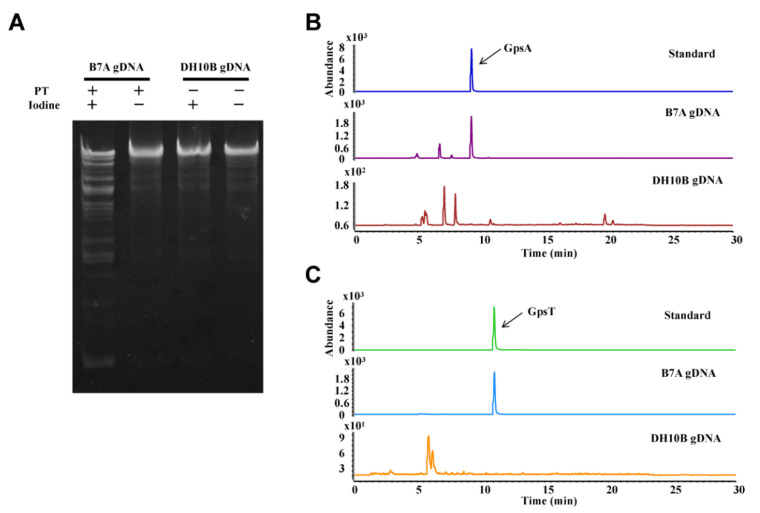
The detection of PT modification. (**A**) ICA analysis of gDNA. Lanes 1–2, genomic DNA from *E. coli* B7A with (+) or without (-) iodine. Lanes 3–4, genomic DNA from *E. coli* DH10B with (+) or without (-) iodine. (**B**) and (**C**) LC-MS/MS analysis of the PT-linked dinucleotides from *E. coli* B7A and from *E. coli* DH10B. The PT-modified dinucleotide (GpsA or GpsT) was used as standards.

**Figure 3 biomolecules-10-01491-f003:**
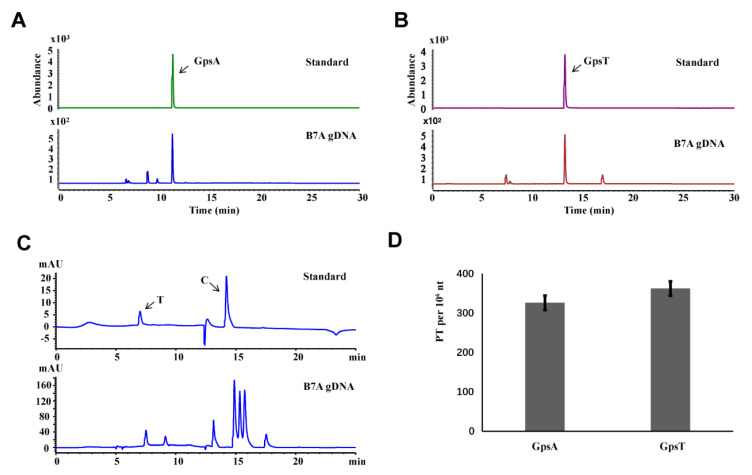
Quantitative analysis of genomic PT modification by LC-MS/MS. (**A**,**B**) LC-MS/MS analysis of the PT-linked dinucleotides from *E. coli* B7A. (**C**) HPLC analysis of gDNA from *E. coli* B7A. (**D**) The number of PT-modified dinucleotides per 10^6^ nt of DNA in *E. coli* B7A. Error bars are calculated as the s.d. of three independent biological replicates.

**Figure 4 biomolecules-10-01491-f004:**
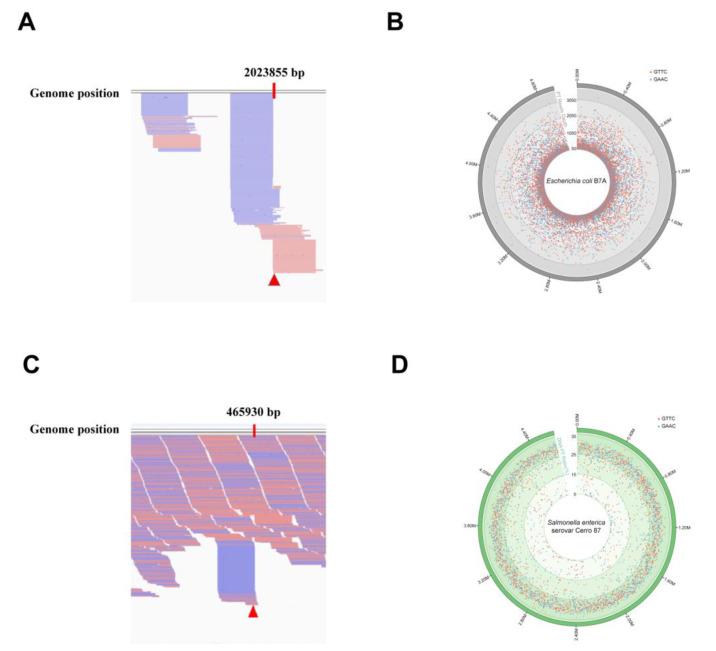
Characterization of PT modifications on whole genomic landscapes by ICDS and PT-IC-Seq. (**A**) The snapshot of genome browser (IGV) representing partially modification sites in *E. coli* B7A genome. The top track shows counts of 5′ ends in the selected region. Blue segments represent reads mapped to reverse strand and red segments represent reads mapped to forward strand. The red triangle marks a PT sites in *E. coli* B7A genome. (**B**) Map of PT modification sites across the *E. coli* B7A genome. From inner to outer circles: 1 to 4 (reads value of modification sites with 50–1050, 1050–2050, 2050–3050, more than 3050): PT sites in GAAC (blue dots) and GTTC (red dots). The dot marks denote all occurrences of the GAAC/GTTC sequence across the genome, and the depth of reads ended at these locations see the Appendix A. (**C**) The snapshot of genome browser representing partially modification site in *Salmonella enterica* serovar Cerro 87 genome. The top track shows counts of 5′ ends in the selected region. Blue segments represent reads mapped to reverse strand and red segments represent reads mapped to forward strand. The red triangle marks a PT site in *Salmonella enterica* serovar Cerro 87 genome. (**D**) PT site mapping across the *Salmonella enterica* serovar Cerro 87 genome. From inner to outer circles: 1 to 3 (PT sites modified with 5–15%, 15–25%, 25–35%): PT sites in GAAC (blue dots) and GTTC (red dots). The dot marks denote all occurrences of the GAAC/GTTC sequence across the genome, and the PT ratios at each site see the Appendix A.

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
