# Peer review of "Development of Methods Derived from Iodine-Induced Specific Cleavage for Identification and Quantitation of DNA Phosphorothioate Modifications"

_biomolecules, 2020, doi:10.3390/biom10111491_

Round 1
Reviewer 1 Report
In this paper the authors declare to “ provide here a comprehensive description…..of a series of approaches for analyzing DNA phosphorothioate modifications…”, all iodine-induced cleavage based approach!
I found very interesting when they inform that the iodine solution should be prepared fresh every time (448) but I didn’t understand what is the RT-IC-Seq (453) method,…. which integrate the integration..?? reported again at line 465 …the ICDS nor the RT-IC-Seq..
It is a bit confusing for me because RT could be misinterpreted as Real Time and in the recent literature there is a paper title “Novel Iodine-induced Cleavage Real-time PCR Assay for Accurate Quantification of Phosphorothioate Modified Sites in Bacterial DNA” which is not cited in this paper.
In my opinion, a comprehensive description of methods for PT modifications analysis should include also the IC-qPCR assay.
Author Response
In this paper the authors declare to “provide here a comprehensive description…..of a series of approaches for analyzing DNA phosphorothioate modifications…”, all iodine-induced cleavage based approach!
I found very interesting when they inform that the iodine solution should be prepared fresh every time (448) but I didn’t understand what is the RT-IC-Seq (453) method,…. which integrate the integration..?? reported again at line 465 …the ICDS nor the RT-IC-Seq..
It is a bit confusing for me because RT could be misinterpreted as Real Time and in the recent literature there is a paper title “Novel Iodine-induced Cleavage Real-time PCR Assay for Accurate Quantification of Phosphorothioate Modified Sites in Bacterial DNA” which is not cited in this paper.
In my opinion, a comprehensive description of methods for PT modifications analysis should include also the IC-qPCR assay.
Thanks very much for the comments and good suggestions. It is well known that iodine is easy to volatilize and decompose, so its concentration becomes lower over time. When its concentration is lower than the threshold, the cutting efficiency will become lower, which would affect the accuracy of experiment. Therefore, using a fresh iodine solution every time was necessary to ensure the accuracy of the results. Deep sequencing of iodine-induced cleavage at PT (ICDS) was developed to map PT
locations in bacterial genomes based on adaptation of high-throughput next generation sequencing technology. In the ICDS approach, iodine reagent was introduced to cleave DNA at PT modification sites, and then ligated to an adaptor with specific index sequence for enriching the DNA fragments with PT modifications by PCR amplification. Since the PCR amplification step will result in enriched amplicons with PT modifications, ICDS sequencing approach can only be used for identifying PT sites but not for quantification. Iodine induced cleavage-based PT sequencing
(PT-IC-seq) was then developed to quantitatively determine the PT modification percentage at each site without enrichment process. The PT modified motifs would be cleaved and should be presented as the reads ends in the sequencing output. The unmodified motifs would not be cleaved and should be present in the internal locations of DNA fragments. The frequency of PT modification can be determined by the ratio of a sequence reads at the end versus internal, which ICDS sequencing could not do. Moreover, the sites with lower modification frequency missing in ICDS could also be detected by PT-IC-seq with high sequencing depth. In order to make it easy to understand, we also revised the statement in the discussion (edited in red font).
However, when the PT modification occur on a single stranded, the treatment of iodine couldn’t introduce double-strand break at modified sites. So, neither the ICDS nor the PT-IC-Seq method can be used to analyze single-stranded PT modifications.
Besides, we are sorry for the mistakes depiction “RT” at line 453 and 465, we have corrected “RT-IC-Seq” to “PT-IC-Seq” in the revised manuscript. Meanwhile, we have cited the paper titled “Novel iodine-induced cleavage real-time PCR assay for accurate quantification of phosphorothioate modified sites in bacterial DNA” and revised the manuscript with simply depiction of this method (edited in red font) in the revised manuscript. Because the comprehensive description of IC-qPCR assay has been made in this article and it is easy to follow accordingly, we did not describe the method in our manuscript. Besides, new references has been added and the number of several references was also revised edited in red.
Reviewer 2 Report
The manuscript describes a series of methods derived from DNA phosphorthioate (PT) specific iodine-induced cleavage: an iodine-induced cleavage assay (ICA), a deep sequencing of iodine-induced cleavage at PT site (ICDS) and an iodine-induced cleavage PT sequencing (PT-IC-Seq), for the investigation of PT modifications in bacterial genome. The authors showed rapid detection of PT modification status in the genome by use of ICA. Moreover, to characterize the PT modification on whole genomic landscapes, the ICDS technology was successfully applied and PT-IC-Seq technology was also developed.
These results provide time-saving and less labor consuming methods for both of qualitative and quantitative analysis of genomic PT modifications and offer for better understanding the biology of the PT modifications in bacteria.
Therefore, I recommend publication after some revisions of the manuscript.
Revisions and Questions:
- The authors described the map of PT modification sites in whole genome landscapes in Figure 5. How different are the results obtained from different colonies?The answer could show the accuracy of author’s method.
- And how many times did the authors do the same experiment (from same colony) in Figure 5? The answer also shows the accuracy of author’s method.
- The author’s explanation (and discussion) is insufficient in each experiment. For example, P10, l375- and Figure 4: the authors described the results of Figure 4D at first. The authors should explain what experiments they have done and what they have achieved from Figure 4A, B and C and then describe about Figure 4D; P11, l396- and Figure 5: the authors described the results about ICDS assay in Figure 5A, B and PT-IC-Seq assay in Figure 5C, D. However, the impact of the both methods was not shown. It is necessary to compare the accuracy and/or convenience with the conventional methods.
- P9, l367- and Figure 3: These experiments are just standard curves for LC-MS/MS. In my opinion, these experiments should be supplementary information.
Minor corrections
P10, l377: please correct “106” to “106” (6 should be a superscript).
Author Response
The manuscript describes a series of methods derived from DNA phosphorthioate (PT) specific iodine-induced cleavage: an iodine-induced cleavage assay (ICA), a deep sequencing of iodine-induced cleavage at PT site (ICDS) and an iodine-induced cleavage PT sequencing (PT-IC-Seq), for the investigation of PT modifications in bacterial genome. The authors showed rapid detection of PT modification status in the genome by use of ICA. Moreover, to characterize the PT modification on whole genomic landscapes, the ICDS technology was successfully applied and PT-IC-Seq technology was also developed.
These results provide time-saving and less labor consuming methods for both of qualitative and quantitative analysis of genomic PT modifications and offer for better understanding the biology of the PT modifications in bacteria.
Therefore, I recommend publication after some revisions of the manuscript.
We really appreciate the reviewer’s comments and suggestions.
Revisions and Questions:
The authors described the map of PT modification sites in whole genome landscapes in Figure 5. How different are the results obtained from different colonies?The answer could show the accuracy of author’s method.
And how many times did the authors do the same experiment (from same colony) in Figure 5? The answer also shows the accuracy of author’s method.
The author’s explanation (and discussion) is insufficient in each experiment. For example, P10, l375- and Figure 4: the authors described the results of Figure 4D at first. The authors should explain what experiments they have done and what they have achieved from Figure 4A, B and C and then describe about Figure 4D; P11, l396- and Figure 5: the authors described the results about ICDS assay in Figure 5A, B and PT-IC-Seq assay in Figure 5C, D. However, the impact of the both methods was not shown. It is necessary to compare the accuracy and/or convenience with the conventional methods.
P9, l367- and Figure 3: These experiments are just standard curves for LC-MS/MS. In my opinion, these experiments should be supplementary information.
Thank you for your questions. According to the heterogeneity of PT modification characterized by PT-IC-Seq in the bacterial population, the individual GAAC/GTTC sites might be PT modified in some cells, but the modification did not occur in all the cells. This means that the PT modification sites of different strains might change in whole genome landscapes between different colonies. So, each value was the mean based on three measurements of the same colony from three repeated experiments.
Meanwhile, we have revised the manuscript with sufficient depiction and explanation of each experiment (edited in red font), and the comparisons with the conventional methods (Dnd phenotype electrophoresis analysis and LC-MS/MS) have been integrated into the newly revised manuscript. Additionally, the standard curves have been integrated into the newly revised supplementary information as Figure S1.
Minor corrections
P10, l377: please correct “106” to “106” (6 should be a superscript).
Sorry for the mistakes, we have corrected “106” to “106” in the revised manuscript (edited in red font).